# Joint Degeneration in a Mouse Model of Pseudoachondroplasia: ER Stress, Inflammation, and Block of Autophagy

**DOI:** 10.3390/ijms22179239

**Published:** 2021-08-26

**Authors:** Jacqueline T. Hecht, Alka C. Veerisetty, Mohammad G. Hossain, Debabrata Patra, Frankie Chiu, Francoise Coustry, Karen L. Posey

**Affiliations:** 1Department of Pediatrics, McGovern Medical School, The University of Texas Health Science Center at Houston (UTHealth), Houston, TX 77030, USA; Jacqueline.T.Hecht@uth.tmc.edu (J.T.H.); Alka.Veerisetty@uth.tmc.edu (A.C.V.); Mohammad.G.Hossain@uth.tmc.edu (M.G.H.); Frankie.Chiu@uth.tmc.edu (F.C.); Francoise.Coustry@uth.tmc.edu (F.C.); 2School of Dentistry, The University of Texas Health Science Center at Houston (UTHealth), Houston, TX 77030, USA; 3Department of Developmental Biology, Washington University School of Medicine, St. Louis, MO 63110, USA; debabratapatra@wustl.edu

**Keywords:** cartilage oligomeric matrix protein, pseudoachondroplasia, autophagy, ER stress, dwarfism, chondrocyte, articular cartilage, joint degeneration

## Abstract

Pseudoachondroplasia (PSACH), a short limb skeletal dysplasia associated with premature joint degeneration, is caused by misfolding mutations in cartilage oligomeric matrix protein (COMP). Here, we define mutant-COMP-induced stress mechanisms that occur in articular chondrocytes of MT-COMP mice, a murine model of PSACH. The accumulation of mutant-COMP in the ER occurred early in MT-COMP articular chondrocytes and stimulated inflammation (TNFα) at 4 weeks, and articular chondrocyte death increased at 8 weeks while ER stress through CHOP was elevated by 12 weeks. Importantly, blockage of autophagy (pS6), the major mechanism that clears the ER, sustained cellular stress in MT-COMP articular chondrocytes. Degeneration of MT-COMP articular cartilage was similar to that observed in PSACH and was associated with increased MMPs, a family of degradative enzymes. Moreover, chronic cellular stresses stimulated senescence. Senescence-associated secretory phenotype (SASP) may play a role in generating and propagating a pro-degradative environment in the MT-COMP murine joint. The loss of CHOP or resveratrol treatment from birth preserved joint health in MT-COMP mice. Taken together, these results indicate that ER stress/CHOP signaling and autophagy blockage are central to mutant-COMP joint degeneration, and MT-COMP mice joint health can be preserved by decreasing articular chondrocyte stress. Future joint sparing therapeutics for PSACH may include resveratrol.

## 1. Introduction

Cartilage oligomeric matrix protein (COMP) is a large, matricellular protein that mediates a variety of cell–cell and cell–matrix interactions [1,2,3,4,5,6,7]. COMP interacts with many extracellular matrix (ECM) proteins, including, but not limited to, collagens types I, II, IX, XII, XIV, matrilin-3, aggrecan, and fibronectin [8,9,10,11,12] and may provide a hub for interaction(s) of collagens with proteoglycans and other ECM proteins [8,9,11]. COMP likely plays a role in the mechanical strength of ECM tissues, as loading increases COMP levels in tendons and aging or overuse decreases abundance [11,13]. Chondrocyte proliferation and chondrogenesis is stimulated by COMP [14,15]. Mutations in COMP cause pseudoachondroplasia (PSACH), a severe dwarfing condition characterized by disproportionate short stature, short limbs, joint laxity, pain, and early onset joint degeneration [2,9,16,17,18,19,20,21,22,23,24,25,26,27,28,29,30,31,32,33,34,35,36,37,38]. PSACH birth parameters are normal, and the first sign of the disorder is decelerating linear growth, starting by the end of the first year, and a waddling gait developing by two years of age [29,32]. Radiographic examination leads to a diagnosis, and the key characteristics are shortening of all the long bones; small abnormal epiphyses; widened and irregular metaphyses; small, underossified capital femoral epiphyses; and platyspondyly [21,29,32,36,37]. While loss of linear growth is the most obvious untoward outcome in PSACH, joint dysfunction and pain are the most debilitating and long-term complications. Pain is significant and begins in childhood, likely from the inflammation, which plays a role in the underlying growth plate chondrocyte pathology [17,21,34,39,40], whereas the pain in adulthood may reflect joint degenerative changes that necessitate hip replacement in a majority of adults [29]. Joint degeneration in PSACH occurs very early in life between the second or third decades, and all joints are affected, especially the hips, elbows, and shoulders [29,37,41,42].

Long before COMP mutations were identified as the cause of PSACH, it was known that growth plate chondrocytes contained massive amounts of material in the rER cisternae. Later studies proved that mutant-COMP does not fold properly and, therefore, is retained in the ER [19,39]. Moreover, mutant-COMP prematurely interacts with binding partners in the ER, forming an ordered matrix composed of types II and IX collagen, matrilin 3 (MATN3), and other ECM proteins, resulting in intracellular protein accumulation [30,31,40]. This material is not degraded efficiently enough to maintain chondrocyte function and fills the cytoplasmic space, becoming toxic to growth plate chondrocytes [2,21,33,35]. To study the cellular mechanism involved in the PSACH pathology, a doxycycline (DOX)-inducible mouse that expresses mutant-D469del-COMP (designated the MT-COMP mouse) in chondrocytes was generated [40]. The MT-COMP mouse mimics the clinical phenotype and chondrocyte PSACH pathology [40,41,42,43]. Studies of the MT-COMP mouse showed that mutant-COMP is retained in the rER of growth plate chondrocytes, stimulating ER stress through the CHOP pathway that in turn activates oxidative and inflammatory processes, initiating a self-perpetuating stress loop involving oxidative stress and inflammation. This stress loop leads to DNA damage, necroptosis, and loss of growth plate chondrocytes [43,44,45]. ER stress and TNFα-driven inflammation increase mTORC1 signaling [42], which represses autophagy, eliminating a critical mechanism needed to clear misfolded proteins and results in intracellular accumulation in growth plate chondrocytes [42]. This molecular mechanistic process explains the deceleration in linear growth associated with PSACH.

In this work, we focused on delineating the effect of mutant-COMP on joint health in our PSACH mouse model and the timing of joint degeneration. Articular and growth plate cartilages serve very different functions, and these tissues mature at different times [46,47]. Growth plate chondrocytes generate a large volume of matrix, driving long-bone growth and, after reaching hypertrophy, chondrocytes die. In contrast, articular chondrocytes synthesize matrix to provide lubrication and a cushion to withstand compressive forces during movement, and they are very long lived. Growth is essential to both fetal and postnatal periods, while withstanding weight bearing compressive forces during ambulation occur postnatally. Both articular and growth plate chondrocytes synthesize different extracellular matrix proteins unique to each tissue. In this study, articular cartilage was evaluated for ER stress, inflammation, autophagy block, proteoglycans, and pro-degradative processes and the timing of each process. While stresses that drive mutant-COMP pathology in growth plate chondrocytes also underlie the articular cartilage pathology, important unique degradative processes were present.

## 2. Results

### 2.1. Mutant-COMP Is Retained in the ER in Adult Articular Chondrocytes

Intracellular retention of mutant-COMP in chondrocytes is a hallmark of PSACH [21,40,41]. Previously, we have shown that mutant-COMP accumulates in the ER of growth plate chondrocytes of MT-COMP mice, beginning at E15 [40]. Figure 1, left panel, shows that human mutant-COMP (red signal) is retained in the ER (green signal) of MT-COMP chondrocytes (yellow merge) at 4, 8, 12, 16, and 20 weeks (Figure 1F,J,N,R,V). In contrast, there is no detectable mutant-COMP in control C57BL\6 mice (Figure 1A,E,I,M,Q,U).

### 2.2. ER Stress, Inflammation, Matrix Degradation, Autophagy Repression, Senescence, and Chondrocyte Death Are Present in MT-COMP Articular Chondrocytes

In growth plate chondrocytes, the intracellular accumulation of mutant-COMP stimulates a complex pathological process. This involves activation of the ER stress through the CHOP pathway, which in turn activates oxidative and inflammatory processes that exacerbates ER stress, causing an over activation of mTORC1 signaling blocking autophagy, and ultimately chondrocyte death [43,44]. As shown in Figure 1, CHOP expression in the MT-COMP articular cartilage is observed at 12, 16, and 20 weeks of age (CHOP panel-P, T, X), while it is absent in the controls (C, G, K, O, P, S, W). CHOP is a pro-apoptotic transcription factor that stimulates cell death when ER stress is unresolved [48]. The presence of CHOP in articular chondrocytes at 12 weeks demonstrates that ER stress is delayed by 8 weeks after the intracellular accumulation of mutant-COMP, which begins at 4 weeks.

Because inflammatory processes involving pro-inflammatory cytokines, IL-1β and TNFα, play a significant role in the growth plate, these cytokines were investigated in the articular cartilage (IL-1β data not shown). Shown in Figure 2, TNFα was evident in the deep zone of articular cartilage at 2 weeks of age and was observed in all zones from 4–20 weeks, compared to little or no signal in the controls, and IL-1β showed a very similar expression pattern. TNFα and IL-1β promote the synthesis of matrix metalloproteinases (MMPs), a family of degradative enzymes that cleave collagens and proteoglycans in the extracellular matrix [49]. MMP13 is synthesized by articular chondrocytes and plays an important role in the ECM degradation of articular cartilage associated with osteoarthritis (OA) [49,50,51]. MMP13 immunostaining was increased in MT-COMP mice with negligible staining at 8 weeks and a strong signal from 12–20 weeks, compared to the control articular cartilage, which showed minimal immunostaining (Figure 2). Consistent with these findings, MMP (2, −3, −9, and −13) activity measured at 12 weeks by MMPSense was significantly higher in MT-COMP knees (4.26 × 10^8^ ± 6.52 × 10^7^) than in the controls (3.28 × 10^8^ ± 1.88 × 10^7^) (*p* < 0.034911) (data not shown). These findings show that joint degeneration-associated cytokines (TNFα beginning at 4 weeks) and elevated MMP13 immunostaining were associated with mutant-COMP accumulation in articular chondrocytes in MT-COMP mice.

Autophagy is repressed in both OA articular chondrocytes [52] and MT-COMP growth plate chondrocytes [42], therefore, MT-COMP articular chondrocytes were assessed for the presence of phosphorylated S6 ribosomal protein (pS6); pS6 is an established readout for mTORC1 signaling that regulates synthesis of cellular components and inhibits autophagy [53]. As shown in Figure 2, pS6 signal was increased in MT-COMP articular chondrocytes, beginning at 12 weeks, indicating autophagy suppression. In addition to autophagy blockage, articular chondrocyte death increased, beginning at 8 weeks. (Figure 3), and the loss of these chondrocytes may impact matrix synthesis. Moreover, the increase in CHOP expression precedes the peak in articular chondrocyte cell death at 16 weeks, suggesting that CHOP contributes to chondrocyte death in MT-COMP mice (Figure 1 CHOP panel and Figure 3).

Senescence is known to play a role in OA joint degeneration [54,55] and was evaluated in MT-COMP mice using the senescence marker p16 INK4a. As shown in Figure 3, p16 INK4a was observed in MT-COMP articular chondrocytes from 16–20 weeks. (with minimal staining at 12 weeks.), consistent with the timing of joint degeneration in MT-COMP mice (Figure 4) [54,55], as well as overlapping with the timing of expression of pS6 (Figure 2). This suggests that MT-COMP articular chondrocytes experience cellular stress, which corresponds to senescent articular chondrocytes by 16 weeks, a finding typically associated with aging and not with relatively young adult mice [56,57,58].

### 2.3. MT-COMP Mice Show Signs of Pain in Early Adulthood

In mice, the presence of pain is measured by changes in specific behaviors [59]. One such behavior is changes in voluntary running. The mouse is housed in a cage with a wheel and the number of rotations are recorded to measure voluntary running based on the assumption that pain with physical activity will reduce the number of wheel rotations [60]. Figure 5A shows that MT-COMP mice ran ≈50% less than the controls from 20 to 24 weeks of age, suggesting that MT-COMP mice are less motivated to run, presumably due to pain with ambulation.

Alterations of gait can indicate the presence of pain because these alteration are a natural attempt to reduce stress on limbs associated with pain during ambulation [61]. The DigiGait system is a treadmill that measures multiple gait parameters and can uncover subtle changes in gait. Gait was evaluated at 16 weeks with mice running at 30 cm/s 15° downhill. As shown in Figure 5B, several gait indices that suggest pain were altered, including: % brake stance; % propel stance; stance width, and hind limb shared stance time. MT-COMP mice spent more time braking and less time in the propel phase of stance. A longer duration in braking stance may indicate more precise control and distribution of load to reduce peak loading [62]. Importantly, hind limb shared stance time and % shared stance time were increased in MT-COMP mice compared to the controls (Figure 5B). Hind limb shared stance time is the amount of time that both limbs are in contact with the surface. Shared stance time increases with joint pain because distribution of body weight over both limbs reduces pain [63]. Overall, these measures of gait disturbance suggest that MT-COMP mice have a wider gait for stability, with care taken when placing paws on the belt and more time with both hind limbs on the belt to minimize stress on the limbs, which are all suggestive of pain.

### 2.4. Joint Degeneration in MT-COMP Mice Is Validated by OA Scoring

OA scoring was used to detect and quantify early degenerative changes in the proteoglycan content of articular cartilage, synovitis, and bone/cartilage. No differences were detected prior to 16 weeks of age. MT-COMP femur proteoglycan content was less, and there was a trend toward higher scores for bone/cartilage damage and total score at 16 weeks. (Table 1). A total score of 5.7 was significantly higher in 20 week MT-COMP mice compared to 2.56 in the controls (with a maximum score of 12). This score included the cartilage/bone damage score of 0.9 in 20 week MT-COMP mice compared to 0.11 in the controls and a synovitis score of 1.8 compared to 0.89, respectively (Table 1). These findings show that joint degeneration occurs in MT-COMP mice by 20 weeks, much earlier than in the C57BL\6 control background strain, where it occurs after a year [64], which is again consistent with early joint degeneration observed in PSACH.

### 2.5. Prevention of Joint Degeneration in MT-COMP Mice with Resveratrol Treatment or Ablation of CHOP

Previously, we have shown that resveratrol treatment reduces ER stress by promoting autophagy that clears accumulated mutant-COMP from the ER in growth plate chondrocytes of MT-COMP mice [65]. Resveratrol relieved the ER protein accumulation and reduced chondrocyte death, restoring proliferation and supporting limb growth in juvenile MT-COMP mice [65]. Based on these results, resveratrol was administered to determine whether it could prevent and/or reduce joint degeneration in MT-COMP mice. As shown in Figure 6, resveratrol treatment from birth to 20 weeks dramatically reduced the loss of proteoglycans in the articular cartilage of adult mice (Figure 6A–D) and decreases total OA score from 5.70 ± 2.14 (MT-COMP) to 1.3 ± 1.55 (MT-COMP +resveratrol *** *p* < 0.0005) similar to controls 2.56 ± 2.15 (data not shown). Importantly, resveratrol treatment prevented the accumulation of mutant-COMP in the ER of articular chondrocytes (Figure 6E–H) reduced ER stress (Figure 6M–P), and substantially reduced articular chondrocyte death (Figure 6I–L) and TNFα inflammation (Figure 6Q–T). Consistent with normalization of proteoglycans in the articular cartilage of MT-COMP mice treated with resveratrol, MMP-13 signal was markedly reduced (Figure 6Y–B’). Resveratrol reduced multiple mutant-COMP pathologies, including inflammation and ER stress, and importantly block autophagy, which allows chondrocytes to clear misfolded protein and prevent cell death. Similarly, genetic ablation of CHOP was used to interrupt the ER stress signaling pathway that breaks down the pathological loop between ER stress, inflammation, and oxidative stress. MT-COMP/CHOP^−/−^ articular cartilage at 20 weeks was healthy (Figure 6D) and MMP-13 was minimal (Figure 6B’). The loss of CHOP considerably reduced accumulation of mutant-COMP in articular chondrocytes (Figure 6H) and the number of TUNEL positive chondrocytes (Figure 6L).

## 3. Discussion

Using our MT-COMP mouse model of PSACH, we found that retention of mutant-COMP in the ER of articular chondrocytes induces and drives a CHOP-dependent ER stress pathologic loop involving multiple inflammatory processes. Persistent inflammation drives autophagy blockage and, ultimately, chondrocyte death. The presence of senescence and a degenerative environment likely adversely impacts surrounding tissue, including synovium and subchondral bone. The results of this study show that MT-COMP mice undergo premature joint degeneration similar to the premature joint degeneration in PSACH, providing a model system for studying nonsurgical joint sparing therapeutics.

Multiple cellular stresses were observed in MT-COMP articular chondrocytes, including ER stress (CHOP), inflammation (TNFα), ECM degradative enzymes (MMP-13), block of autophagy (pS6), and senescence (p16 INK4a). Importantly, these stresses were not observed in MT-COMP mice in the absence of DOX (Appendix A). TNFα stimulates excessive mTORC1 signaling (indicated by pS6), consequently blocking autophagy and preventing mutant-COMP from being cleared from the ER through macroautophagy (referred to as autophagy). The autophagy blockade means that the ER cannot be cleared and the pathology is perpetuated without therapeutic intervention. Moreover, elevated mTORC1 signaling maintains protein synthesis, and this counteracts repression of translation mediated by the unfolded protein response to assist with clearance of the ER.

Aging and/or cellular stress stimulates senescence [57,66], and senescent articular chondrocytes were observed in the articular cartilage of MT-COMP mice at 16 weeks. The multiple chronic stresses that occur in MT-COMP articular chondrocytes likely stimulate senescence similar to that observed in OA [56]. Based on our ER-stress mechanistic model, we expect that inflammation, ER stress, MMP expression, autophagy blockage, senescence, and chondrocyte death will each contribute to articular cartilage erosion [42,43,45,67,68,69,70]. Unique to the MT-COMP articular chondrocytes is the presence of the well-known degradative enzyme MMP-13 and senescence, both of which are associated with OA in humans and animal models [56,71]. These findings suggest that mutant-COMP joint degeneration shares some cellular pathology with OA.

The timing of mutant-COMP joint degeneration is distinct from the mutant-COMP growth plate pathology. In articular chondrocytes, mutant-COMP retention and inflammation start at 4 weeks, and ER stress, blockage of autophagy, and chondrocyte death are seen between 8 and 12 weeks (summarized in Table 2). In contrast, mutant-COMP retention in the growth plate is seen prenatally with inflammation starting postnatally at 2 weeks, peaking between 3–4 weeks, and chondrocyte death is significantly increased by 4 weeks [39,47,48]. These tissues serve very different functions, with growth plate driving linear growth, from birth to 10 weeks, and the articular cartilage needed to cushion mechanical forces that are not required in early life (birth to 3 weeks) when ambulation is limited. Articular cartilages primarily absorb and distribute mechanical forces so these mechanical stresses are not transmitted to the bone. In contrast, the growth plate is a niche for the maturation of growth plate chondrocytes that eventually generate copious amounts of ECM that will be calcified and turned into bone. These functional differences may explain the novel mutant-COMP articular chondrocyte pathology of matrix degradation, senescence, and premature joint degeneration in MT-COMP mice.

MT-COMP mice undergo premature joint degeneration far earlier than the background strain (C57BL\6) that develop joint degeneration beginning at 1 year of age. While there is limited information about the temporal development of joint degeneration in PSACH, natural history studies show that there is premature joint degeneration in PSACH, starting in mid- to late teenage years [32]. Based on this information, we posit that the MT-COMP mouse is a good model system for understanding the PSACH/mutant-COMP pathologies.

The essential role of CHOP in the mutant-COMP joint degenerative process is illustrated by the alleviation of articular chondrocyte stress in MT-COMP/CHOP^−/−^mice. Notably, the MT-COMP/CHOP^−/−^mice still express mutant-COMP, but in the absence of CHOP the mutant-COMP pathology is diminished. Similarly, resveratrol treatment from birth to 20 weeks dampens the mutant-COMP pathologies of ER stress (CHOP), inflammation (TNFα), block of autophagy (pS6), and senescence (p16 INK4a). This reduction of ER stress and inflammation interrupts the pathological loop between ER, oxidative stress, and inflammation. Moreover, resveratrol restoration of autophagy allows clearance of mutant-COMP from the ER of articular chondrocytes. These proof-of-principle findings show that the articular cartilage of MT-COMP mice can be preserved if intervention occurs prior to a point of no return and opens new therapeutic avenues for PSACH.

## 4. Materials and Methods

### 4.1. Bigenic Mice

The MT-COMP mice used in these and previously described experiments contain the pTRE-COMP (coding sequence of human COMP + FLAG tag driven by the tetracycline responsive element promoter) and pTET-On-Col II (rtTA coding sequence driven by a type II collagen promoter) [39,47,53]. DOX (500 ng/mL) was administered to mice at birth, through mother’s milk, to collection, in their drinking water. This study complied with the Guide for the Care and Use of Laboratory Animals, eighth edition (ISBN-10, 0-309-15396-4) and was approved by the Animal Welfare Committee at the University of Texas Medical School at Houston and complies with NIH guidelines.

### 4.2. Generation of CHOP Null Bigenic Mice

CHOP null mice were procured from Jackson Laboratories and mated with MT-COMP bigenic mice to obtain a strain expressing MT-COMP in a CHOP null background (MT-COMP/CHOP^−/−^), as used in previous experiments [43,48]. Genotypes of the CHOP null mice were verified using CHOP-specific primers [48].

### 4.3. Immunohistochemistry

Hind limbs from male and female MT-COMP and C57BL\6 control mice were collected and articular cartilage analyzed, as previously described [40,42,43]. Briefly, the limbs were fixed in 95% ethanol followed by decalcification in immunocal (StatLab McKinney, TX, USA) for 1 week, and pepsin (1 mg/mL in 0.1N HCl) was used for antigen retrieval for immunostaining with antibodies for human COMP (Thermofisher, Waltham, MA, USA; MA1-20221, 1:100), CHOP (Santa Cruz Dallas, TX, USA; SC-575; 1:100), interleukin 1 (IL-1) (Abcam Cambridge, United Kingdom; ab7632, 1:200), tumor necrosis factor α (TNFα) (Abcam, Cambridge, UK; ab6671, 1:200), PDI (Santa Cruz Dallas, TX, USA; SC-20132, 1:100), p16 INK4a—(Abcam Cambridge, United Kingdom; ab189034, 1:200), pS6 (Cell Signaling Technology 2215S rabbit polyclonal, 1:200), and MMP-13 (Abcam Cambridge, United Kingdom: ab39012, 1:50). Species specific biotinylated secondary antibodies were used for 1 hr at RT. Sagittal sections of the same thickness (5 um) were then washed and incubated with streptavidin horseradish peroxidase (HRP), and DAB was used as chromogen. The sections were dehydrated and mounted with cytoseal 60 (Thermofisher, Waltham, MA, USA) and then visualized under a BX51 inverted microscope (Olympus America, Center Valley, PA, USA). Limbs were fixed in 10% wt/vol formalin for terminal deoxynucleotidyl transferase–mediated deoxyuridine triphosphate-biotin nick end labeling (TUNEL) staining. For proteoglycan stains, samples were deparaffinized and hydrated in distilled water and stained with safranin-O (Spectrum Chemical, New Brunswick, NJ, USA, 477-73-6) according to the manufacturer’s protocol. Immunostaining was performed on 10 animals in each group.

### 4.4. Gait Analysis

Gait of male mice was analyzed using a DigiGait treadmill system (Mouse Specifics Inc., Boston, MA, USA) following the protocol described for collagen-induced OA [40]. Previously, it has been shown that running during gait analysis was necessary to uncover subtle changes in ambulatory function that are associated with OA [63]. Briefly, video camera images of mice running through a transparent belt were captured and analyzed. At least 3 sec of downhill 15° running continuous strides were used to calculate and measure gait parameters with DigiGait software version 12. Mice were run for 30 cm/sec and gait analysis was performed in the animal behavioral testing room, and assessments were performed by software so that blinding was not necessary. DigiGait recommends 3 mice per group to measure ethanol induced ataxia or running speed (https://mousespecifics.com/digitgait-protocols/, accessed on 22 May 2021). Based on this information, 5 mice/group were analyzed because OA gait dysfunction is subtler to detect compared to ethanol induced ataxia. ANOVA analysis was performed, followed by pairwise comparisons with Kruskal–Wallis and post hoc Dunn’s analysis. All mice were moved to the behavior room at 8:00 a.m. and DigiGait analysis occurred between 9:00 a.m.–12:30 p.m. Due to the dwarfing phenotype associated with expression of mutant-COMP in mice [45,46,47], only gait parameters not influenced by size were considered. Changes in bone shape (metaphyseal flaring) [45] cannot be accounted for in gait analysis, and this a limitation of this study.

### 4.5. OA Scoring

OA scoring was performed on 10 different 5 um sagittal sections (from individual mice). To ensure sections were from the same area of the joint; only sections that contained both menisci were scored. While OARSI scoring covers a wide range of OA pathology, in this study, OA scoring was modified to optimize evaluation of early OA pathology. Four areas, synovium, bone/cartilage, tibial and femoral articular cartilage, were scored from 0 to 3 on each safranin-O stained section. A score of 0 indicated normal or no damage, 1 = mild damage, 2 = moderate damage, and 3 = denotes severe damage. Synovitis, bone/cartilage damage, and proteoglycan of the tibia and femur were scored individually, and all scores were summed with maximal damage being associated with a score of 12. Synovitis was defined as: mild—increase in thickness of synovial lining and increase in stromal area, moderate—increase in stromal density, or severe—thickening of synovial lining with further increase of stromal cellular density. Bone/cartilage damage was defined as: normal—surface was smooth, mild—minor erosion of the surface, moderate—presence of remodeling with minor erosion, or severe—major erosion. Proteoglycans of the articular cartilage of the tibia and femur was classified as: normal—if staining was even through to the subchondral bone, mild—when staining was thinned, moderate—thinning of proteoglycan stained layer and absence of staining in some areas, or severe—widespread loss of proteoglycan staining. Ten mice per experimental group were used for each time point, providing 80–90% power to detect a minimal difference of 2 or 3 units. All scoring was performed blindly. Section depth, thickness, fixation, and decalcification conditions were all identical for all limbs analyzed. The Kruskal–Wallis test was used to evaluate distribution of OA scores across 6 experiment groups, with Post-hoc Dunn’s test comparing MT-COMP to controls.

### 4.6. MMPSense

Mice were treated with depilatory cream to remove hair prior to imaging. MMP 680 reagent was injected into the tail vein, and animals were imaged 24 h later, as per manufacturer’s instructions, along with an uninjected control (UIC). Males were imaged on an IVIS Spectrum In Vivo Imaging System (PerkinElmer; Waltham, MA, USA) (https://www.perkinelmer.com/lab-solutions/resources/docs/APP_Protocol_MMPSense%20680.pdf, accessed on 22 May 2021). MMP 680 is an optically silent substrate that fluoresces when cleaved by MMP-2, -3, -9, and -13. MMPSense signal was assessed blindly by positioning a circle of a standard size around the knee (in all samples), and radiance efficiency was generated from IVIS software. Six mice were included per group, and the system had the power to detect a difference of 30% or greater. Mann–Whitney U test was used to evaluate MMP activity in control and MT-COMP mice.

## Figures and Tables

**Figure 1 ijms-22-09239-f001:**
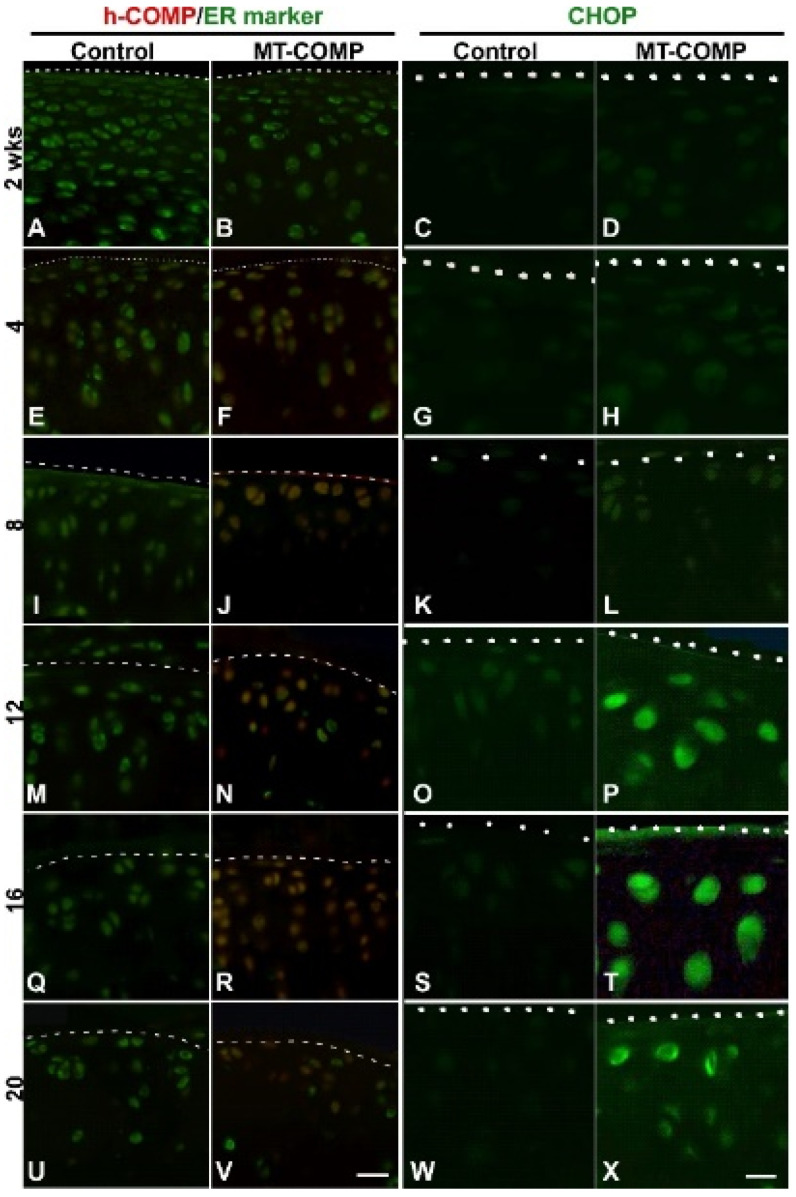
Retention of human mutant-COMP in ER of MT-COMP articular chondrocytes at 4 weeks, with CHOP expression at 12 weeks. All mice were administered DOX from birth to collection at 2, 4, 8, 12, 16, and 20 weeks. Dotted line marks the edge of articular cartilage. Control and MT-COMP tibial articular cartilages were immunostained using human COMP-specific antibodies (red) and protein disulfide isomerase (PDI) (green), an ER marker (**A**,**B**,**E**,**F**,**I**,**J**,**M**,**N**,**Q**,**R**,**U**,**V**). Mutant-COMP was expressed and retained in the ER of articular chondrocytes of MT-COMP at 4 weeks (**F**,**J**,**N**,**R**,**V**) but not in the controls (**A**,**E**,**I**,**M**,**Q**,**U**). CHOP immunostaining is shown in **C**,**D**,**G**,**H**,**K**,**L**,**O**,**P**,**S**,**T**,**W**,**X**. CHOP was present in MT-COMP articular chondrocytes at 12, 16, and 20 weeks (**P**,**T**,**X**) and absent in the controls (**C**,**G**,**K**,**O**,**S**,**W**). (**A**,**B**,**E**,**F**,**I**,**J**,**M**,**N**,**Q**,**R**,**U**,**V**) Bar = 100 µm; (**C**,**D**,**G**,**H**,**K**,**L**,**O**,**P**,**S**,**T**,**W**,**X**) Bar = 50 µm.

**Figure 2 ijms-22-09239-f002:**
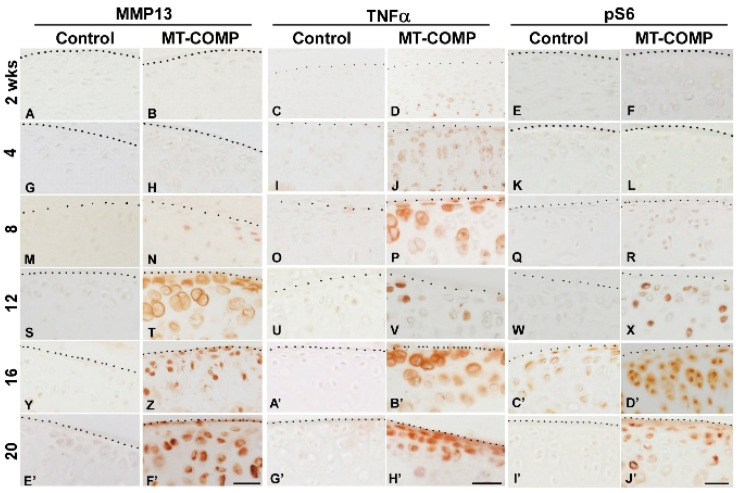
Degradation, inflammation, and autophagy blockage is increased in MT-COMP articular cartilage/chondrocytes. All mice were administered DOX from birth to collection at 2, 4, 8, 12, 16, and 20 weeks. Dotted line marks the edge of articular cartilage. Control and MT-COMP tibial articular cartilage were immunostained (brown) using MMP13 (brown **A**,**B**,**G**,**H**,**M**,**N**,**S**,**T**,**Y**,**Z**,**E’**,**F’**),TNFα (brown **C**,**D**,**I**,**J**,**O**,**P**,**U**,**V**,**A’**,**B’**,**G’**,**H’**), or pS6 (brown **E**,**F**,**K**,**L**,**Q**,**R**,**W**,**X**,**C’**,**D’**,**I’**,**J’**). Control mice show no MMP-13 or TNFα signal and minimal pS6. In contrast, the MT-COMP articular chondrocytes show minimal MMP-13 at 8 weeks, which is increased by 12 weeks; TNFα expression is found in the deep zone of the articular cartilage at 2 weeks of age and in all three layers at 4, 8, 12, 16, and 20 weeks; pS6 signaling is seen at 12, 16, and 20 weeks. Bar = 100 µm.

**Figure 3 ijms-22-09239-f003:**
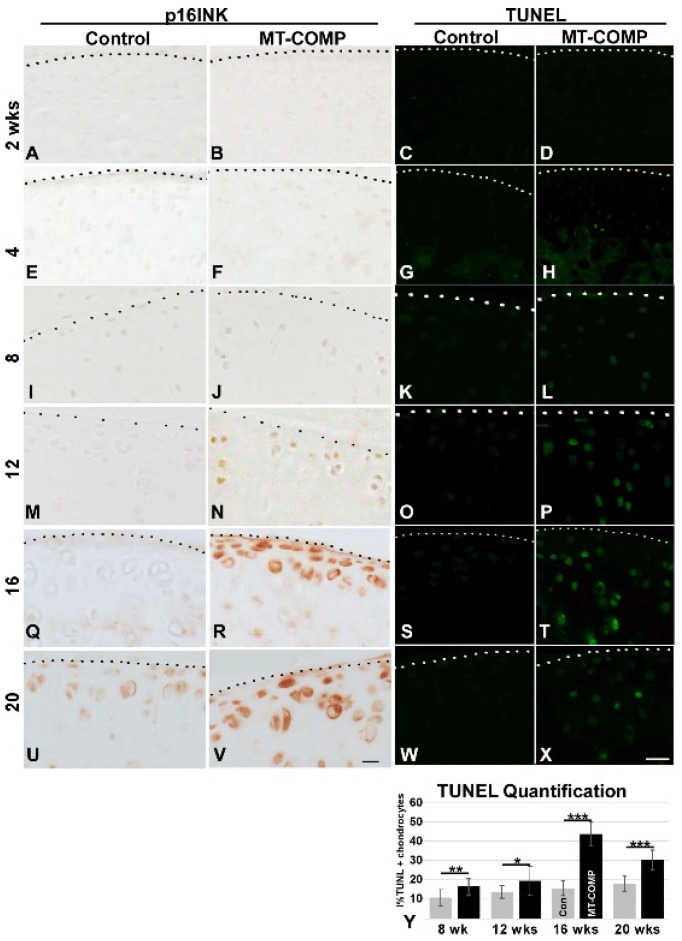
Articular chondrocyte senescence and death in MT-COMP mice. All mice were administered DOX from birth to collection at 2, 4, 8, 12, 16, and 20 weeks. Dotted line marks the edge of articular cartilage. Control and MT-COMP tibial articular cartilage were immunostained using p16 INK4a antibodies (brown **A**,**B**,**E**,**F**,**I**,**J**,**M**,**N**,**Q**,**R**,**U**,**V**). p16 INK4a expression is present from 12–20 weeks (**N**,**R**,**V**). TUNEL staining (green) is shown in **C**,**D**,**G**,**H**,**K**,**L**,**O**,**P**,**S**,**T**,**W**,**X**. The MT-COMP mice show numerous TUNEL positive articular chondrocytes (**L**,**P**,**T**,**X**) compared to the controls (**C**,**G**,**K**,**O**,**S**,**W**). Percent TUNEL positive MT-COMP articular chondrocytes are shown at 8, 12, 16, and 20 weeks. MT-COMP TUNEL staining is significantly different than the controls * *p* < 0.05; ** *p* < 0.005; *** *p* < 0.0005 (**Y**). Sections from the hind limb articular cartilage of at least 10 animals per group were stained with TUNEL. Bar = 100 µm.

**Figure 4 ijms-22-09239-f004:**
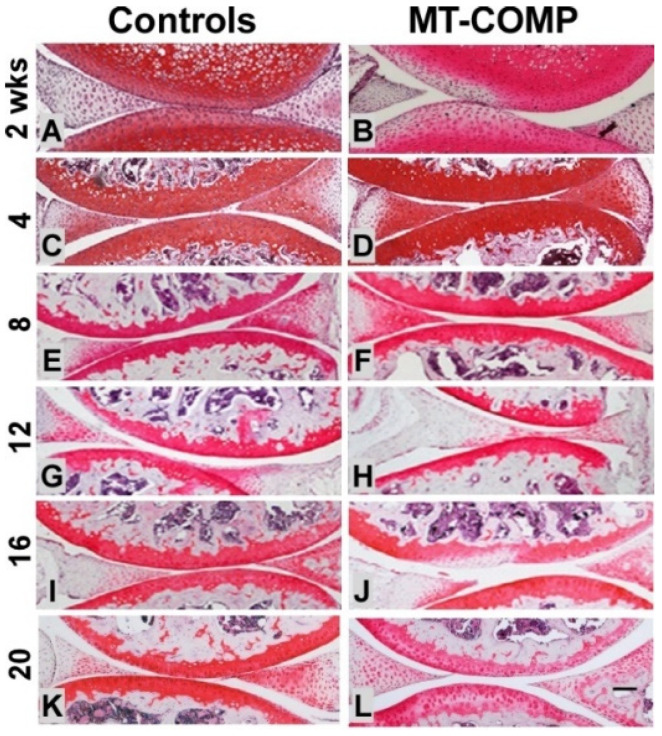
Reduced proteoglycans in MT-COMP mice by 16 weeks. All mice were administered DOX from birth to collection at 2, 4, 8, 12, 16, and 20 weeks. Safranin O staining (red/pink) of the control (**A**,**C**,**E**,**G**,**I**,**K**) and MT-COMP (**B**,**D**,**F**,**H**,**J**,**L**) at 2, 4, 8, 12, 16, and 20 weeks. An abundant and rich proteoglycan layer was found in the control articular cartilage (**A**,**C**,**E**,**G**,**I**,**K**). The proteoglycan layer in MT-COMP mice was similar to the controls at 4 and 8 weeks. (**D**,**F**) but was diminished in select areas at 16 weeks (**J**) and progressed to a more generalized loss at 20 weeks (**L**). Bar = 500 µm.

**Figure 5 ijms-22-09239-f005:**
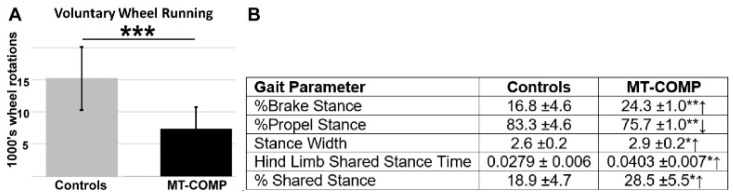
Voluntary running and gait indices. All mice were administered DOX from birth to collection. Voluntary running from 20–24 weeks (**A**) and gait indices (**B**) were measured at 16 weeks. MT-COMP mice ran approximately 50% less than the controls (N = 6/group). Select gait parameters associated with pain are shown and were significantly different (↑,↓) in MT-COMP mice compared to the controls (N = 6/group). MT-COMP mice had increased (↑) % brake stance, stance width, hind limb shared stance time, and % shared stance while % propel stance was decreased (↓). % Brake Stance = the percentage of the stance spent in braking; % Propel Stance = the percentage of the stance spent in propulsion; Stance Width = the perpendicular distance between the centroids of either set of axial paws during peak stance; Hind Limb Shared Stance Time = length of time both hind paws contact belt; % Shared Stance = the percentage of the stance spent with both hind paws in contact with the belt. * *p* < 0.05; ** *p* < 0.005; *** *p* < 0.0005.

**Figure 6 ijms-22-09239-f006:**
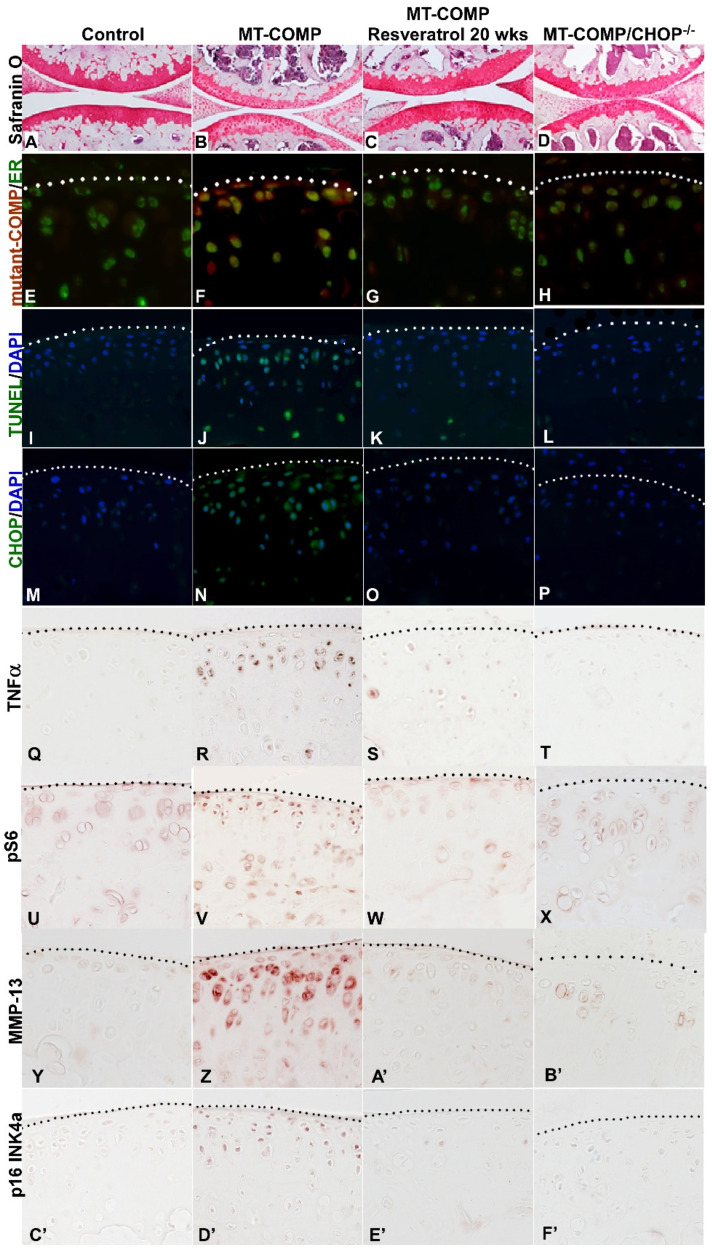
**Articular cartilage of MT-COMP mice is preserved with resveratrol treatment or CHOP ablation.** DOX was administered to all mice and resveratrol treatment to one group of MT-COMP mice (**C**,**G**,**K**,**O**,**S**,**W**,**A’**,**E’**) from birth to 20 weeks. Dotted line marks the edge of articular cartilage. All articular cartilages were stained with safranin O (red/pink **A**–**D**) or immunostained using human COMP-specific antibodies (red, **E**–**H**) and protein disulfide isomerase (PDI), an ER marker (green, **E**–**H**), or TUNEL (green, **I**–**L**) and DAPI (blue nuclei, **I**–**L**), CHOP (green, **M**–**P** ER stress marker) and DAPI (blue, nuclei **M**–**P**), TNFα (brown **Q**–**T**), pS6 (brown **U**–**X**), MMP13 (brown **Y**–**B’**), or p16 INK4a (brown **C’**–**F’**). Safranin O staining shows that MT-COMP articular cartilage has less proteoglycans (**B**) compared to the control (**A**), and both resveratrol treated and ablation of CHOP normalizes proteoglycan content (**C**,**D**). TUNEL positive MT-COMP articular chondrocytes in (**J**) are more numerous than in the control (**I**), resveratrol treated (**K**), or MT-COMP/CHOP^−/−^ (**L**). TNFα inflammation is decreased in resveratrol treated (**S**), or MT-COMP/CHOP^−/−^ (**T**) articular chondrocytes compared to MT-COMP (R). mTORC1 signal activity (pS6) is detected in MT-COMP mice (**V**) compared to the controls (**U**), resveratrol treated MT-COMP (**W**), or MT-COMP/CHOP^−/−^ (**X**) mice. Increased MMP-13 is present in MT-COMP (**Z**) articular cartilage compared to the controls (**Y**), resveratrol treated (**A’**), or MT-COMP/CHOP^−/−^ (**B’**) mice. Senescent articular chondrocytes (p16 INK4a) were observed in MT-COMP mice (**D’**), whereas a minimal p16 INK4a signal is seen in the controls (**C’**), resveratrol treated (**E’**), or MT-COMP/CHOP^−/−^ (**F’**). Bar = 100 µm.

**Table 1 ijms-22-09239-t001:** OA score for MT-COMP and control joints.

Age	Genotype	Pro. Femur	Pro. Tibia	Bone/Cartilage	Synovitis	Total
4 weeks	Control	0.78 ± 0.63	0.56 ± 0.50	0.11 ± 0.31	0.33 ± 0.47	1.78 ± 1.13
	MT-COMP	0.70 ± 0.64	0.30 ± 0.46	0.40 ± 0.66	1.0 ± 0.89	2.4 ± 1.85
	*p* value	-	-	-	-	-
	Control	0.80 ± 0.60	0.60 ± 0.66	0.10 ± 0.30	0.40 ± 0.66	1.90 ± 1.76
8 weeks	MT-COMP	1.3 ± 0.90	0.63 ± 0.14	0.40 ± 0.49	0.60 ± 0.92	2.28 ± 0.11
	*p* value	-	-	-	-	-
12 weeks	Control	0.77 ± 0.32	0.57 ± 0.26	0.30 ± 0.20	0.52 ± 0.30	1.48 ± 1.06
	MT-COM	1.10 ± 0.83	1.20 ± 1.10	0.90 ± 0.94	1.00 ± 0.89	4.20 ± 3.25
	*p* value	-	-	-	-	-
16 weeks	Control	1.00 ± 0.63	0.80 ± 0.60	0.50 ± 0.50	0.90 ± 0.83	3.20 ± 2.27
	MT-COMP	1.60 ± 0.49	1.10 ± 0.54	1.10 ± 0.83	1.50 ± 0.81	5.30 ± 2.19
	*p* value	**0.0372 ***	-	0.0798	-	0.0614
20 weeks	Control	1.00 ± 0.78	0.56 ± 0.65	0.11 ± 0.32	0.89 ± 0.83	2.56 ± 2.15
	MT-COMP	1.80 ± 0.87	1.20 ± 0.60	0.90 ± 0.70	1.80 ± 0.60	5.70 ± 2.14
	*p* value	0.0683	0.0542	**0.0091 ****	**0.0218 ***	**0.0133 ***

Each category is: Pro. Femur = proteoglycan content in femur articular cartilage; Pro. Tibia = proteoglycan content in tibial articular cartilage. Bone/Cartilage and Synovitis are scored from 0–3, with a maximum total of 12. The average in each category ± standard deviation is presented, with significant *p*-values, * *p* < 0.05 and ** *p* < 0.005, in bold text, and trending *p*-values < 0.10 are also shown.

**Table 2 ijms-22-09239-t002:** Timing of articular chondrocyte pathology.

Mutant-COMP Pathology	2	4	8	12	16	20 Weeks
Mutant-COMP intracellular retention		X	X	X	X	X
Proteoglycan loss (Safranin O)					X	X
ER stress (CHOP)				X	X	X
Chondrocyte death (TUNEL)			X	X	X	X
Inflammation (TNFα)	*	X	X	X	X	X
Autophagy block (pS6)			*	X	X	X
Enzymatic degradation (MMP-13)			*	X	X	X
Senescent chondrocytes (p16 INK4a)				*	X	X

DOX administered from birth to collection at 2, 4, 8, 12, 16, and 20 weeks. Mutant-COMP intracellular retention, proteoglycan loss, ER stress, TNFα inflammation, autophagy block, senescent chondrocytes, degradation enzyme MMP-13, and chondrocyte death are present (X) and begin between 4–16 weeks. * = some immunosignal.

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
