# Peer review of "Joint Degeneration in a Mouse Model of Pseudoachondroplasia: ER Stress, Inflammation, and Block of Autophagy"

_ijms, 2021, doi:10.3390/ijms22179239_

Round 1
Reviewer 1 Report
Thank you for sending me this very interesting article on joint degeneration in a model of pseudoachondroplasia. The authors use histology, immunostaining, and gait analysis to study the effect of mutant COMP on articular chondrocytes and joint degeneration in an inducible cell specific mutant mouse line. They demonstrate that treatment with a ER stress relieving drug is able to ameliorate the damage caused by the presence of the mutant protein. This is an interesting finding, with some potential future clinical applications. There are however a few issues with the paper that would prevent me from recommending it for publication as it currently stands
I have the following concerns:
Major:
- You discuss and have previously shown the effect on growth plate chondrocytes in MT-COMP mice. In this paper you focus on the effects in articular chondrocytes and relate it to the OA features of pseudoachondroplasia. I’m concerned that it is difficult to separate these two effects in this model. You say the mutant mice model the clinical phenotype of pseudoachondroplasia. This is a disease of the skeleton where patients are known to have short stature, abnormal shaped bones, and an abnormal gait. Many OA features and indicators will be affected by changes in limb size and shape, but you have not told the reader if there are similar skeletal changes present in the model. Is there a difference in limb length or shape, whole mouse length or weight, etc? How have you accounted for any differences in your analyses? This limitation should certainly be acknowledged in the paper.
- I therefore have concerns about how telling the gait analysis performed is. You are claiming the changes are related specifically to indicators of pain and OA, but could easily simply being a result of the underlying phenotype induced. Abnormal gait is a characteristic feature of pseudoachondroplasia, and normally appears in toddlers, long before OA occurs. I would like to see more basic information about the limbs/mice, and any adjustments you made to account for this issue.
- Not enough information has been given about the adapted OARSI scoring performed. You say you analysed 10 sections from each mouse, but do not indicate the orientation of the sections, the thickness of the sections, or the distance between sections. I would recommend reading Glasson et al 2010 Osteoarthritis and Cartilage for advice. They suggest 4-6 µm sections taken 80 µm apart through the entire articular surface. I also think the text explaining how they calculated the scores for each mouse could be clearer as I found it difficult to know precisely what was done.
- Controls are not adequately explained. It was not clear to me from the legends or methods precisely what the control mice used were. I’m assuming in most cases they were WT mice also treated with DOX for the same length as the MT-COMP mice. This really needs to be specified, with the background of the mice, so the reader can be sure they are similar. If this is the case then the supplemental figure is also inadequate. It shows that in the absence of DOX the MT-COMP mice and Control mice do not show the markers of chondrocyte stress seen in the presence of DOX. However, no age is given. Many of these were only found to be expressed in later weeks, without knowing the age of these mice this figure is not helpful. I would be interested to know if the MT-COMP mice display the other phenotypes of pseudoachondroplasia in the absence of DOX. Do they have normal size and shape of limbs? What about the CHOP ablation double mutants/resveratrol treated mice, is there an effect on limb size there?
Minor:
- Typo in Figure 6 first sentence after the title. Makes it impossible to understand what was meant.
- p3 line 123 the IL-1β data is not shown. But you haven’t said what it showed. Ideally this would be included this in supplementary, with an explanation of what it shows in the text.
- More information about your sections is needed in most figures. What is the orientation? What portion of the joint is being shown? This is vital on sections showing Safranin O staining.
- Sex of the mice can make a difference in OA development. Male mice tend to develop more severe OA in surgical models. This is less likely to be relevant here, but the sex of the mice in each group should be shown.
Author Response
Reviewer 1 Major concerns:
- You discuss and have previously shown the effect on growth plate chondrocytes in MT-COMP mice. In this paper you focus on the effects in articular chondrocytes and relate it to the OA features of pseudoachondroplasia. I’m concerned that it is difficult to separate these two effects in this model. You say the mutant mice model the clinical phenotype of pseudoachondroplasia. This is a disease of the skeleton where patients are known to have short stature, abnormal shaped bones, and an abnormal gait. Many OA features and indicators will be affected by changes in limb size and shape, but you have not told the reader if there are similar skeletal changes present in the model. Is there a difference in limb length or shape, whole mouse length or weight, etc? How have you accounted for any differences in your analyses? This limitation should certainly be acknowledged in the paper.
In earlier work, we have detailed the changes on limb shape and length induced by mutant-COMP expression in mice. Briefly, metaphyseal flaring 1 is present at 4 weeks of age along with hind limb shortening 1-4 that persists at least until 12 weeks of age 2, 4. Resveratrol treatment from birth to 4 weeks of age increases limb length with approximately a 50% rescue 5. In our analysis, we have accounted for these differences in size by analyzing gait parameters that are percentages that are not impacted by size, however we are unable to account for changes in shape in gait analysis and this limitation has been acknowledged in the methods section.
- I therefore have concerns about how telling the gait analysis performed is. You are claiming the changes are related specifically to indicators of pain and OA, but could easily simply being a result of the underlying phenotype induced. Abnormal gait is a characteristic feature of pseudoachondroplasia, and normally appears in toddlers, long before OA occurs. I would like to see more basic information about the limbs/mice, and any adjustments you made to account for this issue.
In our analysis, we have accounted for these differences in size by analyzing gait parameters that are percentages that are not impacted by size. We are unable to account for changes in shape with analysis. Young children with PSACH have a characteristic waddling gait, however this is not noted in the MT-COMP mice and this difference may be due to differences between biped and quadruped gait. Gait changes are not apparent at 12 weeks of age when proteoglycan loss is not evident suggesting that gait changes at 16 weeks of age maybe associated with proteoglycan loss and/or senescent chondrocytes. Moreover, voluntary running is dramatically reduced in MT-COMP mice from 20-24 weeks suggesting a reluctance to exercise.
- Not enough information has been given about the adapted OARSI scoring performed. You say you analysed 10 sections from each mouse, but do not indicate the orientation of the sections, the thickness of the sections, or the distance between sections. I would recommend reading Glasson et al 2010 Osteoarthritis and Cartilage for advice. They suggest 4-6 µm sections taken 80 µm apart through the entire articular surface. I also think the text explaining how they calculated the scores for each mouse could be clearer as I found it difficult to know precisely what was done.
To score synovitis and ensure the same area of the joint was evaluated for each mouse sections from the center of joint which includes both menisci were evaluated. All sections were the same thickness (5 um) and limbs were sectioned in the sagittal plane. This information is included in the methods section along with a clarification on scoring.
- Controls are not adequately explained. It was not clear to me from the legends or methods precisely what the control mice used were. I’m assuming in most cases they were WT mice also treated with DOX for the same length as the MT-COMP mice. This really needs to be specified, with the background of the mice, so the reader can be sure they are similar. If this is the case then the supplemental figure is also inadequate. It shows that in the absence of DOX the MT-COMP mice and Control mice do not show the markers of chondrocyte stress seen in the presence of DOX. However, no age is given. Many of these were only found to be expressed in later weeks, without knowing the age of these mice this figure is not helpful.
Controls and C57BL\6 wild-type mice treated with DOX in Figures 1-6. Control wild-type mice and MT-COMP mice in Scheme 1 (Supplemental Figure 1) were not administered DOX and this is the appropriate control for this experiment. Figure 6 and Scheme 1 (Supplemental Figure 1) show sections from animals at 20 weeks of age and this information has been added to legends.
- I would be interested to know if the MT-COMP mice display the other phenotypes of pseudoachondroplasia in the absence of DOX. Do they have normal size and shape of limbs?
Limb size and shape has not been evaluated in MT-COMP in the absence of DOX because the without DOX there is no detectable accumulation of mis-folded COMP in the ER of chondrocytes. The association of accumulation of misfolded proteins in growth plate chondrocytes with decreased long bone growth has been demonstrated by expression of a thyroglobulin mutant specifically in chondrocytes leading to a loss of limb growth 6.
- What about the CHOP ablation double mutants/resveratrol treated mice, is there an effect on limb size there?
Resveratrol treatment from birth to 4 weeks of age increases limb length with an approximately a 50% rescue of lost growth 5 and this is mentioned in section 2.5. Limb length of MT-COMP mice with 20 weeks of resveratrol treatment has not been evaluated and is outside of the scope of this work given the focus of this work is the impact of mutant-COMP accumulation on the articular cartilage and much published work describes the growth aspects of the disorder. According to Jackson Labs, CHOP null mice are smaller than wild-type mice and therefore limb length assessments are confounded by this size difference.
Reviewer 1 Minor:
- Typo in Figure 6 first sentence after the title. Makes it impossible to understand what was meant.
Typo has been corrected.
- p3 line 123 the IL-1β data is not shown. But you haven’t said what it showed. Ideally this would be included this in supplementary, with an explanation of what it shows in the text.
IL-1B results are reported on page 3.
- More information about your sections is needed in most figures. What is the orientation? What portion of the joint is being shown? This is vital on sections showing Safranin O staining.
All sections at 5 um and are in the sagittal orientation. To score synovitis sections from the center of joint which includes both menisci were evaluated and this information is included in the methods section.
- Sex of the mice can make a difference in OA development. Male mice tend to develop more severe OA in surgical models. This is less likely to be relevant here, but the sex of the mice in each group should be shown.
Male mice were evaluated in these experiments presented here, however female mice were evaluated for chondrocyte pathology (human COMP-specific antibodies, TUNEL, CHOP, TNFα, pS6, MMP13 and p16 INK4a) which was very similar to that observed in MT-COMP males. Sex information is included in the methods section.

Reviewer 2 Report
Please see attached pdf file for comments.

Author Response
Reviewer 2 Minor:
We agree to all minor edits that Reviewer 2 indicated in the pdf.
Reviewer 2 References:
Reviewer 2 indicated that most references cited were older than 5 years and suggested we update references. Pseudoachondroplasia is a rare disorder with few researchers dedicated to study the disorder and therefore the field does not move forward as fast as many other fields. Consequently, the appropriate references tend to be older than 5 years.
